# Female genital lichen sclerosus is connected with a higher depression rate, decreased sexual quality of life and diminished work productivity

Olga Jabłonowska[1]*, Anna Woźniacka[1], Simona Szkarłat[2], Agnieszka Żebrowska[1]

**1** Dermatology and Venereology Clinic of the Medical University of Lodz, Lodz, Poland, **2** Department of Urology, Hospital of the Ministry of Interior and Administration in Lodz, Lodz, Poland

* olga.jablonowska@gmail.com

**Data Availability Statement:** We stored our data in the open access repository Zenodo. The DOI to access our data is 10.5281/zenodo.7782999.

## Abstract

Female genital lichen sclerosus is an underdiagnosed, distressing, chronic dermatosis affecting the well-being of women. The aim of this retrospective case-control study was to assess whether the disease is connected with work productivity and activity impairment, depression and decreased sexual quality of life. Fifty-one female patients with genital lichen sclerosus and forty-five healthy women were enrolled to the study and filled out an online survey including: Work Productivity and Activity Impairment: General Health (WPAI:GH), Patient Health Questionnaire-9 (PHQ-9) and The Sexual Quality of Life-Female (SQOL-F) questionnaires. The results showed that women with genital lichen sclerosus are at risk of having a diminished work productivity, are more often screened for depression and have a decreased sexual quality of life. The study highlights the importance of a multidisciplinary approach to treating female genital lichen sclerosus.

## Introduction

Lichen sclerosus (LS) is a chronic, distressing skin condition characterised by relapses and remissions. The pathogenesis of LS is still not clear. The symptoms of the disease include itching, burning, sexual dysfunctions, pain during voiding or defecation and may lead to decreased quality of life. The disease is diagnosed in men and women of all ages, with the highest prevalence in postmenopausal women, reaching up to 3% of this population [1]. However, recent studies suggest that the true incidence of LS in younger women is underestimated [2, 3].

The disease usually affects anogenital region, in 15–20% of cases extragenital area is involved [4, 5]. About 6% of patients present only with extragenital lesions [6]. Due to the frequent involvement of genital organs there are some reports about impaired quality of sexual life in LS patients [7–11]. Additionally, skin diseases were associated with increased self-disgust, which to a greater extent was experienced by females [12]. Furthermore, the sexual needs of women and professional career aspects are often neglected. Therefore the aim of this retrospective case-control study was to assess the influence of female genital LS on sexual

**Funding:** The study was funded by the Medical University of Lodz, project no. 503/1-152-01/503-11-002.The funders had no role in study design, data collection and analysis, decision to publish, or preparation of the manuscript.

**Competing interests:** The authors have declared that no competing interests exist.

satisfaction, screen for depressive disorder and determine whether LS is connected with diminished productivity in the workplace.

## Materials and methods

Between January and December 2021 an online survey including: Work Productivity and Activity Impairment: General Health (WPAI:GH), Patient Health Questionnaire-9 (PHQ-9), The Sexual Quality of Life-Female (SQOL-F) questionnaires was conducted [13–15]. Fifty-one adult female patients with genital LS, filled out an online questionnaire and were enrolled in the study group. Also demographic and clinical data: age, place of residence, professional activity, previous and current treatment, was collected. The diagnosis of LS was set clinically by the doctor, based on characteristic features of skin lesions and concomitant symptoms. In a few cases it was additionally confirmed by a skin biopsy. Thirty women declared to be using topical corticosteroids, 4 of whom only milder potency ones, 6 applied calcineurin inhibitors, 16 mentioned using other treatment options (e.g. photodynamic therapy, laser, platelet rich plasma), 3 did not use any form of treatment. The corresponding control group consisted of 45 healthy women with a similar demographic status (p>0.05). Fig 1 The study was exempt from research ethics review by the formal opinion of the Bioethics Committee of the Medical University of Lodz (No. RNN/282/22/KE). Written informed consent was obtained from all participants of the study.

Patient Health Questionnaire-9 (PHQ-9)–a validated screening tool for diagnosing and measuring the severity of depression—was performed in the study and control groups [14]. As a screening method for major depression the cut-off score was established at 10 points [16–18]. Using an alternative 'diagnostic algorithm' major depressive syndrome (MDS) and other depressive syndrome (ODS) can be detected. In MDS five items, whereas in ODS two, three or four items from PHQ-9, have to be scored for 2 points or more. Additionally, to fulfil the alternative criteria of MDS and ODS one of these items should concern depressed mood or anhedonia [14].

Sexual Quality of Life Questionnaire-Female is a psychomteric instrument developed by Symonds et al. which aims to measure the influence of sexual dysfunctions on the quality of women's sexual life [15]. It is composed of 18 questions with a 6-point rating scale each. The total score ranges between 18–108 points. The cut-off indicating adequate sexual quality of life is not determined. The higher the score, the better the quality of sexual life. Four factors concerning four different aspects of sexual life can be distinguished: psychosexual feelings (seven items), sexual and relationship satisfaction (five items), self-worthlessness (three items), sexual repression (three items).

Statistical analysis using the Mann Whitney U test and Chi-squared test was performed. A p-value ≤ 0.05 was considered statistically significant.

## Results

### WPAI:GH

Fifty-one women filled out the WPAI:GH questionnaire. Seven of the participants were not professionally active. There were 3 pensioners, 2 students and 2 women who did not work due to unknown reasons. The influence of lichen sclerosus on work activity was further analysed in the remaining 44 employed women. During the last 7 days the disease caused excused absence from work in 7 people (16%). One person did not work due to depression and was qualified for rehabilitation allowance. Other 6 women (15%) were on sick leave which lasted from 1 to 24 hours during the last 7 days, with the median duration of 10.5 hours. Moreover 20 women (45%) during past week took the vacation leave which lasted for 2–56 hours, reaching 18.8

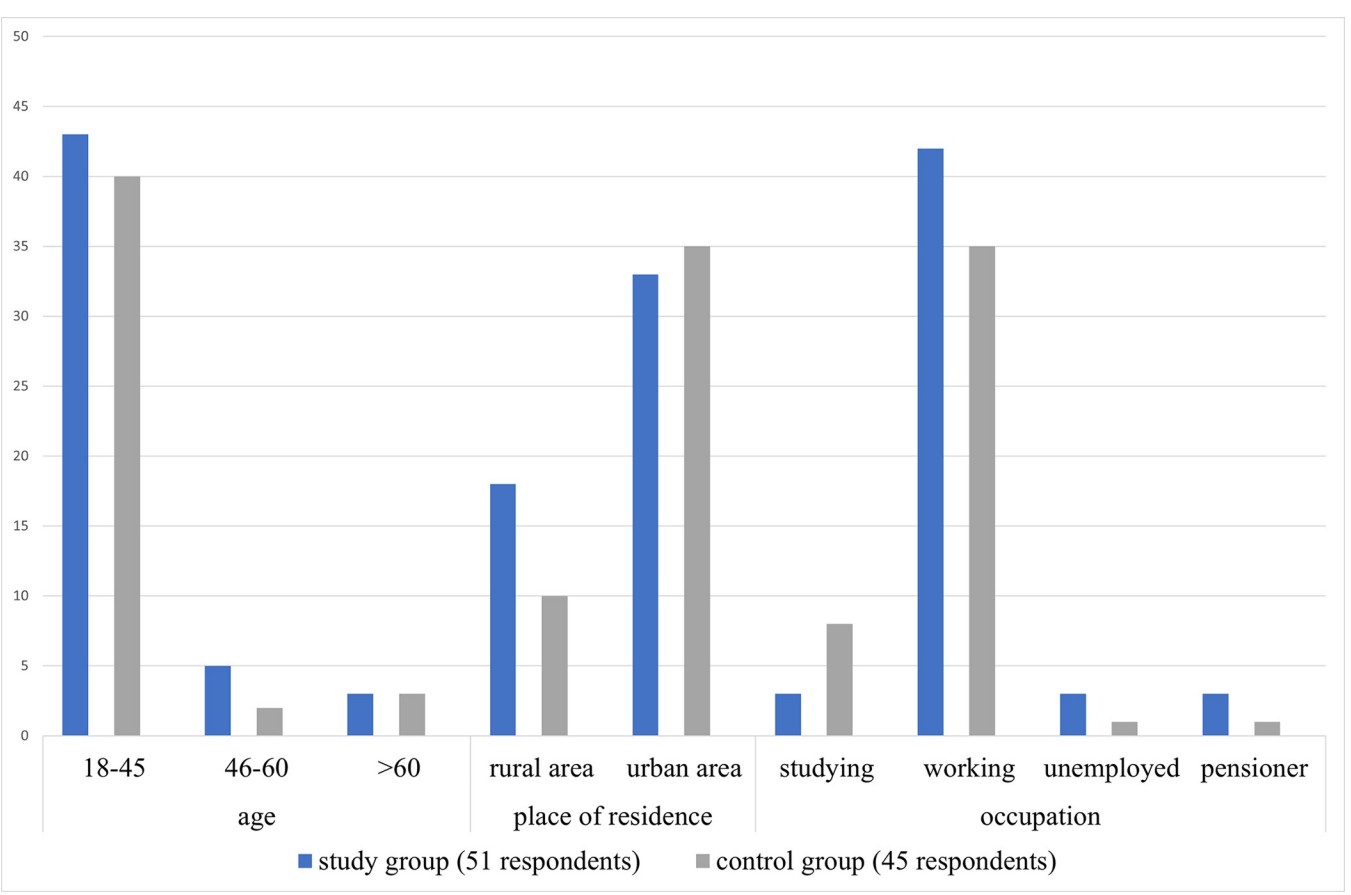

**Fig 1. Group characteristics.** Characteristics of the study and control group (p>0.05).

hours on average. The working time during past week ranged from 0 to 55 hours, approximately 25.5 h. Twenty eight women (64%) claimed that health problems impaired theirs work (≥2 points). The influence of the disease on work activity was assessed as moderate (5-7/10 points) in 10 individuals (23%) and significant (8-10/10 points) in 4 women (9%). Fig 2 Regarding the impact of the disease on daily activities 11 women (25%) determined it as moderate (5-7/10 points) and 7 (16%) as significant (8-10/10 points). Mean percentages for absenteeism and presenteeism were 5.32% and 31.35%, respectively. The total work productivity loss was established as 36.67%.

## PHQ-9

The median PHQ-9 score was 10 (IQR = 10) points in the study group and 6 (IQR = 5) points in the control group (p = 0.0006). Twenty seven women from the study group (53%) and 7 (16%) from the control group had a score ≥10 points (p = 0.0001). Using the alternative 'diagnostic algorithm' the results indicated that 15/51 (29%) women with LS and 2/45 (4%) controls could suffer from major depressive syndrome (MDS) (p = 0.0034). Moreover 9/51 (18%) females from the study group and 4/45 (9%) controls fulfilled the criteria of other depressive syndrome (ODS) (p = 0.3408) [14, 19]. The prevalence rate of depressive disorder based on PHQ-9 was altogether 47% in the study group and 13% in the control group (p = 0.0004) Table 1.

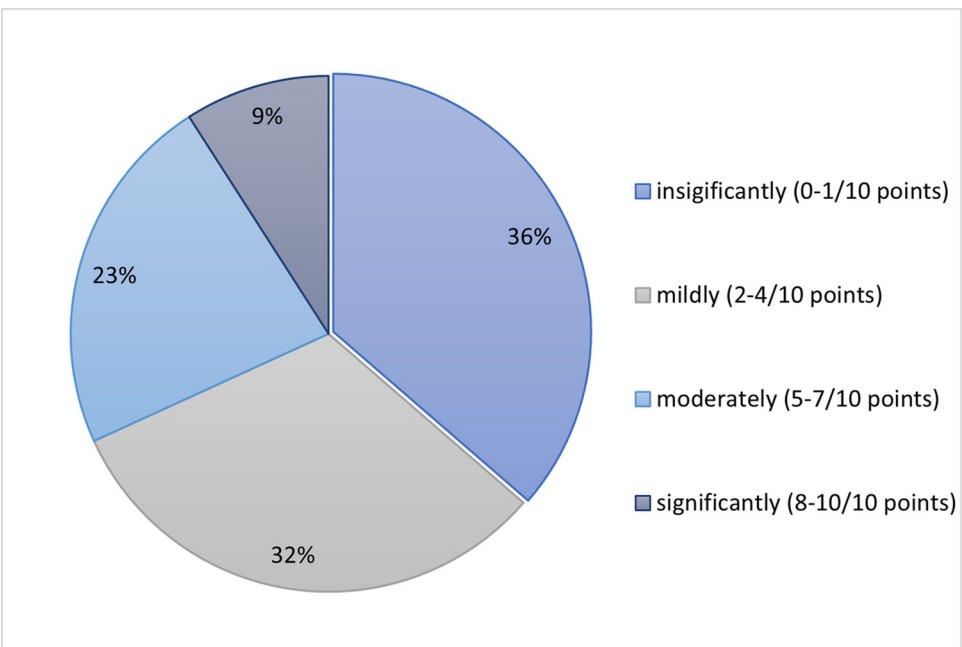

**Fig 2. The influence of LS on work productivity.** The influence of the disease on work productivity of 44 professionally active females with genital LS based on question No. 5 from WPAI:GH questionnaire.

### SQoL-F

In our study 51 female patients with LS and 45 healthy women filled out the online SQoL-F questionnaire. The median of the total score was 47 (IQR = 13) points in the study and 79 (IQR = 13) points in the control group (p = <0.0001). The median score obtained in each of four factors 'psychosexual feelings', 'sexual and relationship satisfaction', 'self-worthlessness', 'sexual repression' was 13, 19, 7, 6 points in women with genital LS and 36, 11, 16, 16 points in controls, respectively (p<0.0001 for all four factors) Fig 3.

## Discussion

It is a well-known fact that chronic diseases may influence the activity of the individual in their life including their professional life [20]. Our study shows that female genital LS is a distressing, chronic dermatosis which not only has a negative impact on the sexual sphere, but also appears to be connected with a higher depression rate and lower productivity at work. There is

**Table 1. The Patient Health Questionnaire-9 –comparison of the results.**

| PHQ-9 | study group | control group | P |
|---|---|---|---|
| **Total Score: Median (IQR)** | 10(10) | 6(5) | 0.0006 |
| **Score ≥ 10** | 27 | 7 | 0.0001 |
| **Major depressive syndrome** | 15 | 2 | 0.0034 |
| **Other depressive syndrome** | 9 | 4 | 0.3408 |
| **Depressive disorder** | 24 | 6 | 0.0004 |

Comparison of the median value of the Patient Health Questionnaire-9 (p = 0.0006), frequency of major depressive syndrome (p = 0.0034), other depressive syndrome (p = 0.3408) and depressive disorder based on Patient Health Questionnaire-9 in the study and control group.

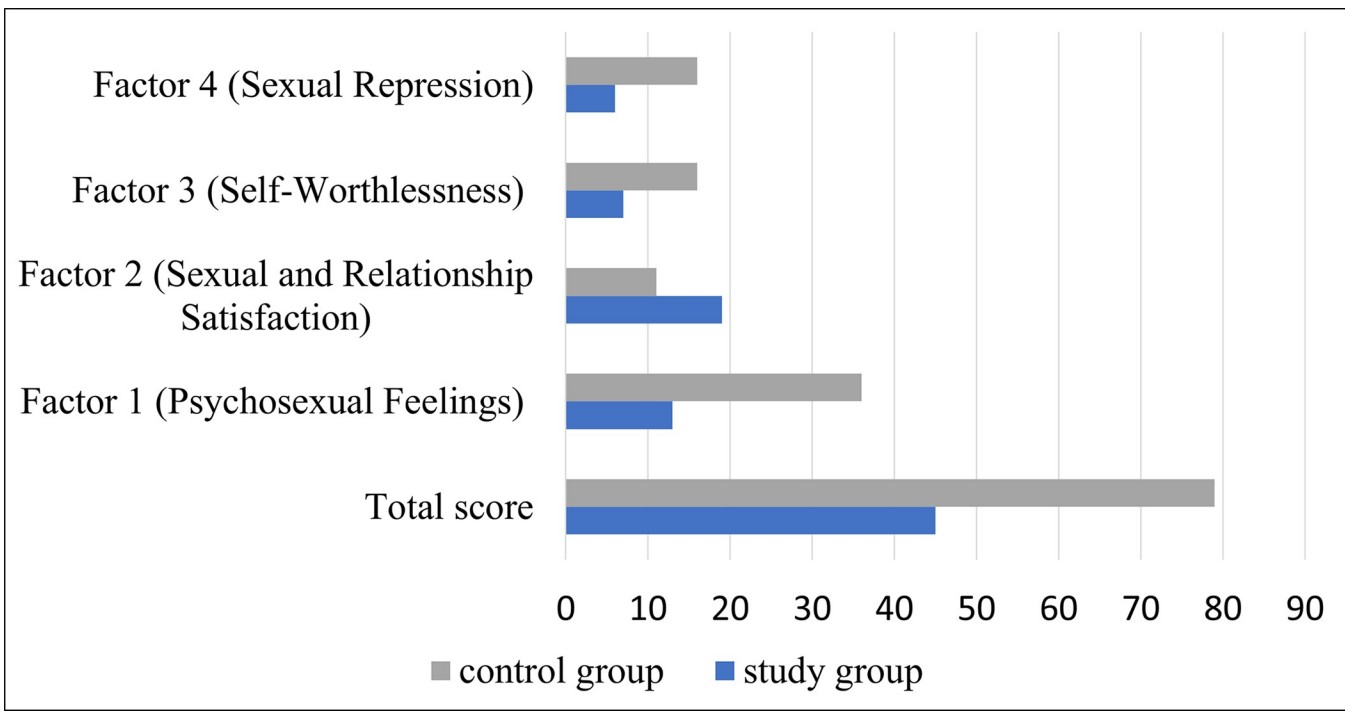

**Fig 3. SQOL-F median; p<0.0001.** Comparison of the median value of the Sexual Quality of Life-Female Questionnaire results in the study and control group (p<0.0001).

some data about diminished sexual quality of life in LS patients, but the effect of the disease on the professional life has not yet been evaluated.

Results from our study indicate that female genital LS exerts a negative impact on work productivity. Similar conclusions were drawn in other research on different chronic diseases: psoriasis, psoriatic arthritis and rheumatoid arthritis [21–23]. Moreover, Villacorta outlined that psoriasis influences economic burden and is also connected with indirect costs caused e.g. by unemployment, absenteeism, presenteeism [22].

Additionally, higher impact of the disease on work was reported in women with psoriatic arthritis and lichen planus than in men [24, 25]. It has been hypothesised that females are more concerned about skin lesions and are prone to experience stigmatization more frequently.

The prevalence rate of depression using PHQ-9 questionnaire has not been determined in the Polish population. However, a study of a representative sample of the general population of Germany using PHQ-9 questionnaire showed that mood disorders were more frequent in women than in men, reaching 4.9% for major depressive syndrome and 7.4% for other depressive syndrome in female population [26]. The prevalence rate of depressive disorder in healthy individuals from our control group was 13%, which is similar to representative sample of the female population of Germany (12.3%). Picardi et al. reported that 18 out of 141 (12.8%) patients with different skin diseases from an Italian dermatological hospital were positively screened for major depression [27]. Similar results were observed in a group of patients with vitiligo [28].

To compare significantly more women with genital LS from our study (47%) fulfilled the criteria of depressive disorder according to PHQ-9 questionnaire. Furthermore, depression was found to be connected with lower sexual quality of life [15]. In a qualitative interview

study vulvar lichen sclerosus was described as a high maintenance disease that caused numerous changes in patients' lives [29]. This illustrates the complexity of the disease and challenges with its management. Another recent study showed an increased self-disgust in patients with dermatological conditions such as, psoriasis, hidradenitis suppurativa and lichen sclerosus [12]. Besides, depression and anxiety were said to correlate significantly with self-disgust [30].

In our study group the median of total SQoL-F score was 47 points, which is significantly lower in comparison with females from the control group and suggests a diminished quality of sexual life in women with genital LS. Women from the study group scored less points in the first, third and fourth factors of SQoL, representing 'psychosexual feelings', 'self-worthlessness', 'sexual repression'. The most notable difference was observed in the first factor associated with females' feelings concerning sexual experiences. It can be simply connected with the physical pain felt during the intercourse due to the skin changes caused by the disease. But also the emotional discomfort associated with the presence of skin lesions and feeling that something is not right with oneself, feeling less attractive and/or less of a woman are likely contributing factors. Interestingly, women with genital LS scored more points than controls in the second factor of SQoL-F which concerns 'sexual and relationship satisfaction'. About three quarters of women from the study group claimed to feel to be able to talk with the partner about sexual matters. It may be presumed that females with genital LS from our study group frequently obtained support from partners and the lower SQoL-F total score was rather not connected with relationship deterioration. Symonds et al. showed that satisfaction with partner relationship was positively associated with SQoL. Conversely, the presence of symptoms such as: difficulty becoming aroused, pain or discomfort, lack of lubrication, taking a long time to become aroused, difficulty achieving orgasm correlated with lower SQoL-F total score [15].

There is also some data about decreased sexual quality of life in patients suffering from alopecia areata and high prevalence of sexual distress in women with psoriasis, especially when the genital area was involved [31]. Moreover, the SQoL-F total score assessed in premenopausal women with diabetes I and II was twofold higher than in our study group [32]. Furthermore, some research showed that women with genital LS had lower genital self-image which was also linked with a higher anxiety level and worse sexual function [33–35]. These data indicate that presence of visible lesions is often perceived by patients as defects and may have a strong influence on emotional and sexual well-being. Qualitative studies concerning vulvar dermatosis showed that there is still a lot od secrecy and stigma around vulvar diseases. Women often complained about shame and loneliness, lack of information and support [29, 36].

Noteworthy, it was reported that psychosexual counselling enhanced sexual functioning of women with genital LS [37]. Also sex education offered to women with endometriosis and patients after myocardial infarction exerted positive influence on their sexual quality of life [38, 39]. Therefore it might be beneficial to actively ask patients with LS about any concerns or fears related to the sexual sphere and consider professional counselling if needed.

## Conclusions

The results of our study showed that not only the sexual quality of life is diminished in women with genital LS, but also the disease may be connected with a higher rate of depression and a negative influence on work. Thus, it is our role as medical practitioners to offer multidisciplinary care to every woman with genital LS because its physical and psychological aspects are strongly related. Work productivity may improve with the better control of the disease and a higher self-esteem.

## Author Contributions

**Conceptualization:** Olga Jabłonowska, Agnieszka Żebrowska.

**Data curation:** Olga Jabłonowska, Simona Szkarłat.

**Formal analysis:** Olga Jabłonowska, Agnieszka Żebrowska.

**Funding acquisition:** Anna Woźniacka.

**Investigation:** Olga Jabłonowska.

**Methodology:** Olga Jabłonowska.

**Project administration:** Agnieszka Żebrowska.

**Resources:** Olga Jabłonowska, Anna Woźniacka.

**Software:** Olga Jabłonowska, Simona Szkarłat.

**Supervision:** Anna Woźniacka, Agnieszka Żebrowska.

**Validation:** Olga Jabłonowska, Agnieszka Żebrowska.

**Visualization:** Olga Jabłonowska, Simona Szkarłat, Agnieszka Żebrowska.

**Writing – original draft:** Olga Jabłonowska.

**Writing – review & editing:** Agnieszka Żebrowska.

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
