## [Decision Letter · Decision Letter 0]

27 Feb 2023

PONE-D-22-29013Female genital lichen sclerosus is connected with a higher depression rate, decreased sexual quality of life and diminished work productivity.PLOS ONE

Dear Dr. Jabłonowska

Thank you for submitting your manuscript to PLOS ONE. After careful consideration, we feel that it has merit but does not fully meet PLOS ONE’s publication criteria as it currently stands. Therefore, we invite you to submit a revised version of the manuscript that addresses the points raised during the review process.

The reviewers have expressed positive comments regarding your article, raising only few concerns. Considering this point, I invite authors to perform the required minor revisions.

We look forward to receiving your revised manuscript.

Kind regards,

Alessandro Favilli, PhD, MD

Academic Editor

PLOS ONE

Journal Requirements:

2. Please ensure that you have specified (1) whether consent was informed and (2) what type you obtained (for instance, written or verbal, and if verbal, how it was documented and witnessed). If your study included minors, state whether you obtained consent from parents or guardians. If the need for consent was waived by the ethics committee, please include this information.

Reviewers' comments:

Reviewer's Responses to Questions

**Comments to the Author**

1. Is the manuscript technically sound, and do the data support the conclusions?

Reviewer #1: Yes

Reviewer #2: Yes

2. Has the statistical analysis been performed appropriately and rigorously? 

Reviewer #1: Yes

Reviewer #2: Yes

3. Have the authors made all data underlying the findings in their manuscript fully available?

Reviewer #1: Yes

Reviewer #2: Yes

4. Is the manuscript presented in an intelligible fashion and written in standard English?

Reviewer #1: Yes

Reviewer #2: Yes

5. Review Comments to the Author

Reviewer #1: The authors have answered all arised questions point by point. The issue is important and the paper is weitten with a clear language. Paper is well revised no further comments. The paper is worth to be published in the journal

Reviewer #2: This paper reports a case-control study on lichen sclerosus and assesses a domain of the disease not in deep studied so far. Additionally, conclusion are in agreement with my own opinion on the disease. Moreover, it is easy to read and understand, reaching the goal of clarity I like in assessing articles.

I have some issues to be pointed out:

1. Criteria for diagnosing LS.

2. Specifying if some treatment for LS have been administered or not at the time of respondents’ answers, as it might affect them.

3. Describe best cases and controls characteristics. I understand you have matched controls, but you have not reported any planned design for doing it. Ideally, time frame in which case and controls have been enrolled would be the same and both cases and controls would came from same population. Finally, case-control studies are usually retrospective. Is this a retrospective study?

4. A structured abstract would be able to transfer the information of a case control study.

I thank you in advance.

Your,

UI

6. PLOS authors have the option to publish the peer review history of their article (what does this mean?). If published, this will include your full peer review and any attached files.

Reviewer #1: **Yes: **Cihan Kaya

Reviewer #2: **Yes: **Ugo Indraccolo

---

## [Author Response · Author response to Decision Letter 0]

29 Mar 2023

Journal Requirements:

1. “Please ensure that your manuscript meets PLOS ONE's style requirements, including those for file naming. The PLOS ONE style templates can be found at 

https://journals.plos.org/plosone/s/file?id=ba62/PLOSOne_formatting_sample_title_authors_affiliations.pdf”

We reviewed the PLOS ONE's style requirements, changed the file naming and saved figure files as “Fig1.tif”, “Fig2.tif”.

2. “Please ensure that you have specified (1) whether consent was informed and (2) what type you obtained (for instance, written or verbal, and if verbal, how it was documented and witnessed). If your study included minors, state whether you obtained consent from parents or guardians. If the need for consent was waived by the ethics committee, please include this information.”

We specified that written informed consent was obtained from all participants of the study. We did not include minors in our study. We elaborate on the ethics committee’s opinion under point 4 below.

3. “We note that you have stated that you will provide repository information for your data at acceptance. Should your manuscript be accepted for publication, we will hold it until you provide the relevant accession numbers or DOIs necessary to access your data. If you wish to make changes to your Data Availability statement, please describe these changes in your cover letter and we will update your Data Availability statement to reflect the information you provide.”

We have stored our data in the open access repository Zenodo. The DOI to access our data is 10.5281/zenodo.7782999

4. “Please include your full ethics statement in the ‘Methods’ section of your manuscript file. In your statement, please include the full name of the IRB or ethics committee who approved or waived your study, as well as whether or not you obtained informed written or verbal consent. If consent was waived for your study, please include this information in your statement as well.”

Information about the ethics committee’s opinion was included in the ‘Methods’ section. The study was exempt from research ethics review based on the formal opinion of the Bioethics Committee of the Medical University of Lodz of 13 December 2022 (No. RNN/282/22/KE). We enclose the original document along with its translation.

5. “Please review your reference list to ensure that it is complete and correct. If you have cited papers that have been retracted, please include the rationale for doing so in the manuscript text, or remove these references and replace them with relevant current references. Any changes to the reference list should be mentioned in the rebuttal letter that accompanies your revised manuscript. If you need to cite a retracted article, indicate the article’s retracted status in the References list and also include a citation and full reference for the retraction notice.”

We reviewed the reference list following PLOS ONE requirements.

6. We also reassessed the status of the women who declared in the Work Productivity and Activity Impairment: General Health (WPAI:GH) questionnaire not to be working. Seven of the participants were not professionally active. There were 3 pensioners, 2 students and 2 women who did not work due to unknown reasons. We revised it in the manuscript.

Reviewer #2:

1. “Criteria for diagnosing LS.”

We explained in the revised version of the paper that the study group included individuals in whom the diagnosis of LS was set clinically by the doctor, based on characteristic features of skin lesions and concomitant symptoms. In a few cases the diagnosis was additionally confirmed by a skin biopsy. 

2. “Specifying if some treatment for LS have been administered or not at the time of respondents’ answers, as it might affect them.”

We specified that 30 women in our study group declared to be using topical corticosteroids, 4 of whom only milder potency ones, 6 applied calcineurin inhibitors, 16 mentioned using other treatment options (e.g. photodynamic therapy, laser, platelet rich plasma) and 3 did not use any form of treatment.

3. “Describe best cases and controls characteristics. I understand you have matched controls, but you have not reported any planned design for doing it. Ideally, time frame in which case and controls have been enrolled would be the same and both cases and controls would came from same population. Finally, case-control studies are usually retrospective. Is this a retrospective study?”

The study is a retrospective case-control study. The process of enrollment to the study and control group took place between January and December 2021. Cases and controls were matched based on gender, age, occupation and place of residence. The characteristics of the study and control group are presented graphically on the chart (Fig 1).

4. “A structured abstract would be able to transfer the information of a case control study.”

We added information about the nature of the study – retrospective case-control study – in the abstract of the paper.

---

## [Editor Report · Decision Letter 1]

13 Apr 2023

Female genital lichen sclerosus is connected with a higher depression rate, decreased sexual quality of life and diminished work productivity.

PONE-D-22-29013R1

Dear Dr. Jabłonowska,

We’re pleased to inform you that your manuscript has been judged scientifically suitable for publication and will be formally accepted for publication once it meets all outstanding technical requirements.

Kind regards,

Alessandro Favilli, PhD, MD

Academic Editor

PLOS ONE
---

## [Editor Report · Acceptance letter]

17 Apr 2023

PONE-D-22-29013R1 

Female genital lichen sclerosus is connected with a higher depression rate, decreased sexual quality of life and diminished work productivity. 

Dear Dr. Jabłonowska:

I'm pleased to inform you that your manuscript has been deemed suitable for publication in PLOS ONE. Congratulations! Your manuscript is now with our production department. 

Kind regards, 

on behalf of

Dr. Alessandro Favilli 

Academic Editor

PLOS ONE